# Higher Immunological Response after BNT162b2 Vaccination among COVID-19 Convalescents—The Data from the Study among Healthcare Workers in an Infectious Diseases Center

**DOI:** 10.3390/vaccines10122158

**Published:** 2022-12-15

**Authors:** Agata Skrzat-Klapaczyńska, Justyna Dominika Kowalska, Marcin Paciorek, Joanna Puła, Carlo Bieńkowski, Dominika Krogulec, Jarosław Stengiel, Agnieszka Pawełczyk, Karol Perlejewski, Sylwia Osuch, Marek Radkowski, Andrzej Horban

**Affiliations:** 1Department of Adults’ Infectious Diseases, Medical University of Warsaw, 02-091 Warsaw, Poland; 2Ward 7th, Hospital for Infectious Diseases, 01-201 Warsaw, Poland; 3PolandHIV Out-Patients Clinic, Hospital for Infectious Diseases, 01-201 Warsaw, Poland; 4Department of Immunopathology of Infectious and Parasitic Diseases, Medical University in Warsaw, 02-091 Warsaw, Poland

**Keywords:** COVID-19 vaccination, healthcare workers, COVID-19 convalescent, immunological response, antibodies

## Abstract

**Introduction:** The BNT162b2 vaccination studies did not specifically focus on groups that were heavily exposed to SARS-CoV-2 infection. Therefore, we aimed to assess the safety and efficacy of the BNT162b2 vaccine among healthcare workers (HCWs). **Methods:** Study participants were recruited from hospital employees who received BNT162b2 vaccination at the Hospital for Infectious Diseases in Warsaw. Blood samples were collected before and after each vaccination dose. At each timepoint, the levels of anti-SARS CoV-2 IgM, anti-n SARS-CoV-2 IgG, and S-RBD antibodies were measured. Data on concomitant diseases and the vaccine’s adverse events (VAE) were collected after each vaccination dose. In the statistical analyses, non-parametric tests were used. **Results:** In total, 170 healthcare workers were included in the analysis. Their median age was 51 years (interquartile range (IQR): 41–60 years); most of them were women (*n* = 137, 80.6%) working in direct contact with patients (*n* = 137, 73.2%); and 46 (27.0%) had concomitant diseases. More than one fifth of subjects had COVID-19 before their first dose of vaccination (*n* = 38, 22.6%). In terms of immunological responses, our investigations showed a high level of efficacy for the BNT162b2 mRNA vaccination as measured by S-RBD antibody concentrations: these were positive in 100% of participants 14 days after the second dose of the vaccine. It was also observed that employees with high S-RBD antibodies (>=433 BAU/mL) were more likely to be COVID-19 convalescents before receiving the first vaccine dose (*p* < 0.001). **Conclusion:** The BNT162b2 vaccine is safe and effective among HCWs. Vaccine adverse events occurred, but serious events were not observed. Moreover, the BNT162b2 vaccine is effective against symptomatic and severe COVID-19—none of the workers that acquired a SARS-CoV-2 infection after vaccination required hospitalization or medical care. We also observed higher immunological responses among COVID-19 convalescents.

## 1. Introduction

The first human cases of the coronavirus disease (COVID-19), caused by SARS-CoV-2, were reported in Wuhan (China) in December 2019 [1]. Since then, there have been over 500 million confirmed COVID-19 cases worldwide, and over 6 million deaths (as of July 2022) [2]. In Poland, over 6 million COVID-19 cases have been diagnosed (up to July 2022), and almost 116,500 people have died [2]. Vaccine development may have been the best method for preventing severe courses of the disease, and it was hoped to reduce the death rate [3]. On 27 December 2020, the National Vaccination Programme was introduced in Poland, providing the BNT162b2 mRNA vaccine (Pfizer-BioNTech, Moguncja, Germany) to high-risk groups [4,5].

Healthcare workers (HCWs), being on the front lines, were deemed as a group with the highest risk of exposure to COVID-19, and a significant number of them were reported to be infected [6]. HCWs were, thus, prioritized for vaccination when COVID-19 vaccines became available in Poland. The reason for this was not only a greater occupational risk of becoming infected, but also the need for healthy HCWs during a critical time, as well as the risk of SARS CoV-2 transmission among family members and the general public [7].

The response to COVID-19 vaccination may be assessed in a few different ways, including the following: the humoral response, the cellular response, and clinical measurement, assessed by the decline in COVID-19 morbidity [8]. However, the BNT162b2 registration studies did not specifically focus on groups that were particularly exposed to SARS-CoV-2 infection, including HCWs [9]. The antibody response and protection of the BNT162b2 vaccine has been established among HCWs previously [10,11,12,13,14,15,16]. Despite this, we extended these findings and focused on HCWs as more highly exposed groups, setting aside the studies on long-term care facility workers.

We, therefore, aimed to assess the safety and efficacy of the BNT162b2 vaccine in a group of employees at the Hospital for Infectious Diseases in Warsaw.

## 2. Methods

Study participants were recruited from the National COVID-19 Vaccination Program provided by a Polish Ministry of Health [17]. The study subjects were all healthcare workers at the time of their vaccination with the BNT162b2 mRNA vaccine (Pfizer-BioNTech, Moguncja, Germany). Blood samples were collected on the day of the first BNT162b2 mRNA vaccination dose and on the day of the second vaccination dose (three weeks after the first dose), and then again at fourteen days and six months after the second vaccination dose. Samples were taken again on the day of the third vaccination dose (nine months after the first vaccination dose), and then again at fourteen days and six months after the third vaccination dose (main group, Table 1). According to the protocol of the study, we distinguished a subgroup which checked the blood more frequently. Thus, in addition to the above blood samples, samples within this group were also collected seven and fourteen days after the day of the first BNT162b2 mRNA vaccination dose, as well as seven days after the day of the second vaccination dose, (Subgroup, Table 1).

The levels of SARS CoV-2 IgM antibodies, SARS-CoV-2 IgG antibodies (against the n-protein, indicative of disease), and S-RBD antibodies (indicative of a response to vaccination) were measured using the MAGLUMI SARS CoV-2 IgM, MAGLUMI SARS CoV-2 IgG, and MAGLUMI SARS CoV-2 S-RBD IgG assays. According to the manufacturer’s information, MAGLUMI^®^ SARS-CoV-2 S-RBD IgG kits are 99.6% specific and 100% sensitive [18]. The kits have been approved for sale in the European Union and have received a CE certificate.

According to laboratory standards, the following antibody levels were interpreted as positive: IgM ≥ 1.10 AU/mL for SARS-CoV-2 antibodies, IgG ≥ 1.00 AU/mL for SARS-CoV-2 antibodies, and ≥4.33 BAU/mL for the S-RBD antibodies. Samples with values over 433 BAU/mL were diluted and measured at a ratio of 1:10 or 1:20 (if necessary), allowing for the extension of the dynamic range of analysis to 8660 BAU/mL.

The COVID-19 cases were monitored in the first eight months only in symptomatic patients (fever, cough, runny nose, general weakness, sore throat, muscle aches, sore throat, and loss of smell and/or taste) and/or under the applicable standard in our infectious diseases center (every healthcare worker was tested by PCR every two weeks). After eight months, standards in our center changed, and only symptomatic subject were monitored by swabbing the nasopharynx. A positive PCR test result for SARS CoV-2 authorized a COVID-19 diagnosis.

Participants were asked to fill in a standardized questionnaire after every vaccination dose; data on concomitant diseases, age, weight, gender, and self-reported vaccine adverse events (VAE) were collected.

Participants with blood samples available on the day specified in the study protocol were included into the current analysis (Table 1).

In the statistical analyses, patients with high and low response rates at six months after the third vaccine dose were compared. A high response was defined as values over 433 BAU/mL based on other studies [19].

Non-parametric tests were used as appropriate: the Wilcoxon test was used to compare dependent numeric variables, while the Fisher exact test and the Chi2 test were used to compare categorical variables. A *p* value of < 0.05 was considered significant, and all tests were two-sided. All analyses were performed using SAS version 9.4 (SAS Institute, Cary, NC, USA).

The study was approved by the Bioethical Committee of the Medical University of Warsaw (Nr KB/2/2021). The study was funded from a research grant issued by the Medical Research Agency (Nr 2021/ABM/COVID19/WUM).

## 3. Results

In total, 202 participants were recruited for the study. Thirty-two study participants were excluded from the analysis, as at least one blood sample was missing from the study scheme. Finally, 170 participants were included in the final analysis. Their median age was 51 years (interquartile range (IQR): 41–60 years). Their median body mass index (BMI) was 25.10 [IQR: 22.68–29.03]. Most of the cohort consisted of women (*n* = 137, 80.6%) who were working directly with patients (*n* = 137, 73.2%). Among the hospital’s employees, 46 (27.0%) had concomitant diseases, of which the three most common were hypertension (*n* = 32, 18.8%), asthma (*n* = 6, 3.5%), and chronic hepatitis (*n* = 3, 1.8%), with immunosuppressive treatment (*n* = 3, 1.8%). More than one fifth of participants had contracted COVID-19 either before being vaccinated (*n* = 38, 22.6%) or after receiving the third BNT162b2 mRNA vaccine dose (*n* = 45, 32.3%); eight (4.7%) persons had contracted COVID-19 more than once. Positive results for SARS-CoV-2 IgM, IgG, and S-RBD antibody assessment among healthcare workers in the Hospital for Infectious Diseases in Warsaw were performed in the table, (Table 2).

Vaccine adverse events (VAE) occurred after every vaccine dose: the first BNT162b2 mRNA dose (*n* = 122, 73.9%), the second BNT162b2 mRNA dose (*n* = 129, 81.5%), and the third BNT162b2 mRNA dose (*n* = 126, 86.3%). However, none of these VAEs were severe, and none required hospitalization. The three most common VAEs after the first dose were as follows: pain at the area of injection (*n* = 109, 66.0%), malaise (*n* = 28, 16.9%), myalgia (*n* = 27, 16.3%), and headache (*n* = 27, 16.6%). After the second BNT162b2 mRNA vaccine dose, these were pain at the area of injection (*n* = 109, 68.9%), malaise (*n* = 67, 42.4%), and myalgia (*n* = 48, 30.4%). After the third BNT162b2 mRNA vaccine dose, these were pain at the area of injection (*n* = 106, 72.6%), malaise (*n* = 57, 39.4%), and headache (*n* = 39, 26.7%).

In addition, we distinguished a group of 110 participants who had the antibody assessment performed 3 weeks after the first vaccine dose. In this group, employees who had high responses to vaccination were younger (41 years [IQR: 34–46]) vs. 51 [IQR: 41–61], *p* = 0.0012), had lower BMI (23.18 [21.45–25.10] vs. 25.64 [IQR: 23.03–29.03], *p* = 0.0296), and were more likely to be diagnosed with COVID-19 before being vaccinated (13/23, 61.9% vs. 3/87, 3.5%, *p* < 0.0001).

## 4. Subgroup Analyses

In total, 49 patients qualified for analyses of the immunological responses’ detailed dynamics. Within this group, seven participants had a high response seven days after the first vaccine dose. Regarding subgroup baseline characteristics, in total, 49 participants were included in the analysis. Their median age was 49.50 years (interquartile range (IQR): 39.50–57.50 years). Their median body mass index (BMI) was 25.32 (IQR: 23.18–29.05). Most of the cohort consisted of women (*n* = 34, 69.39%)who were working directly with patients (*n* = 34, 69.39%).

Employees with a high immunological response were more likely to be COVID-19 convalescents before receiving the first vaccine dose (6/7, 85.7% vs. 1/42, 2.4%, *p* < 0.001). The proportion of adverse events was also comparable, irrespective of the level of serological responses to the first vaccination (Table 3).

The analysis, performed in the subgroup of 49 employees that were tested more often for IgG S-RBD antibody concentrations, revealed that after every vaccine dose, there was a statistically significant increase in S-RBD antibody concentrations (Table 4 and Table 5, Figure 1 and Figure 2).

## 5. Discussion

We conducted a single-center, prospective study to evaluate the immune response to a complete, three-dose BNT162b2 mRNA (Pfizer-BioNTech, Manufacturer: Pfizer, Inc., and BioNTech, Moguncja, Germany) COVID-19 vaccination in 170 healthcare professionals. In addition, we analyzed any vaccine adverse effects that occurred after each vaccination in the study group. The results of our investigations showed a high efficacy of BNT162b2 mRNA vaccination in terms of S-RBD antibody concentrations, which were positive in 100% of participants 14 days after the second dose of the vaccine. Moreover, these concentrations were already highly positive (97%) on the day the second vaccination dose was administered. High S-RBD antibody concentrations were measured in an investigation by Kaneko et al.; they were positive in 98% of participants 14 days after their first vaccination, and 100% positive 7 days after their second dose. The study was conducted among medical workers who received two doses of the vaccine, one on day 0 and the other on day 21 [20].

Similar results showing an adequate humoral response measured by S-RBD antibody concentrations were obtained in a study by Italian authors, who examined the immune response to two doses of the BNT162b2 mRNA vaccine with the same interval of 21 days. The study group consisted of 167 healthcare workers who were proven to be COVID-19 naïve; 100 percent of the recipients had strongly induced antibody production two weeks after the second dose of the vaccine [21]. In a subgroup of our study participants, who were tested more frequently (additional examination of blood samples at 7 and 14 days after the day of the first dose, and 7 days after the day of the second dose of vaccine), a statistically significant increase in the level of antibodies was shown as early as 7 days after the administration of both the second and third doses of the vaccine. An investigation by Kim et al. showed a response to BNT162b2 mRNA vaccination, manifested by the production of S-RBD antibodies 7 days after the second dose of the vaccine in all 283 study participants [22]. Moreover, an analysis by Sahin et al. revealed that the SARS-CoV-2 serum geometric mean for neutralizing antibody concentrations, measured one week after the second dose of the vaccine, was up to 3.3-fold greater when compared to the results of individuals who had recovered from COVID-19 and had not been vaccinated [23]. The high immunogenicity of the vaccine (96–98%) when measured seven days after the second dose was also demonstrated in an analysis by Lusting et al., who examined a large group of medical workers. This work also showed that lower antibody production was independently associated with the male sex, older age, immunosuppression, diabetes, hypertension, heart diseases, and autoimmune diseases [24].

Interestingly, the results of our analysis of the antibody concentrations 7 days after the first dose of the vaccine showed significantly more frequent, very high responses, as measured by S-RBD antibody concentrations in the group of participants who had contracted COVID-19 prior to the initiation of the vaccination course compared to those who were COVID-19 naïve (*p* < 0.0001). Similar conclusions were drawn by the authors of this study by comparing the immune response in convalescents, as well as participants who had not previously suffered from COVID-19, to two doses of the vaccine. The results of this analysis showed significantly higher levels of specific IgG and IgM antibody concentrations in vaccinated COVID-19 convalescents [25].

In an investigation by Zhang et al. examining the production of antibodies in convalescents and COVID-19 naïve participants, the results showed an over twenty-fold increase in the level of antibodies produced against both the SARS-CoV-2 wild-type and the Delta variant in convalescents after one dose of the vaccine, administered 6 months after the onset of COVID-19. Moreover, the measured antibody levels in convalescent patients were greater than in those who were previously COVID-19 naïve, and in those who had been vaccinated with two, instead of one, dose of the same vaccine [26]. Hall et al. in their study, showed that a previous history of COVID-19 was associated with an 84% lower risk of SARS CoV-2 infection [12]. In a study by Tretyn et al., in which, contrastingly, all participants (both convalescent and COVID-19 naïve) had the same vaccination protocol, anti-SARS-CoV-2 IgG antibodies were measured 2 and 3 weeks after the first dose of the vaccine and 1–5 weeks after the second dose of the mRNA vaccine. In all the measurements performed, the anti-SARS-CoV-2 IgG antibody concentration was higher in COVID-19 convalescents compared to the subgroup who had not had a prior SARS-CoV-2 infection [27]. These results are also similar in our study, which showed that in our infectious diseases center, HCWs who had contracted COVID-19 before being vaccinated had a higher immunological response at one week and three weeks after the vaccination. In the study conducted by Spitzer et al. among HCWs at a single center in Israel, it was found that in subjects who had previously been vaccinated with a two-dose series of BNT162b2 vaccination, administration of a booster dose, compared with not receiving one, was associated with a significantly lower rate of SARS-CoV-2 infection [15]. These studies have been supported by other reports which suggested that BNT162b2 is moderately to highly effective in reducing infectivity, not only among vaccinated individuals, but also among unvaccinated adult household members, via preventing infection and through reducing viral shedding [13,14]. Regarding the vaccine adverse events after COVID-19 vaccination, we found it interesting that the proportion of adverse events was comparable irrespective of the level of serological response to the first vaccination, which means that the serological response was not related to the vaccine reaction. This fact may suggest that only individual predisposition is relevant regarding the response to vaccination.

## 6. Conclusions

The BNT162b2 vaccine is safe, VAEs may occur; however, in the study, none of these were severe, and none required hospitalization or medical care.The proportion of adverse events was comparable irrespective of the level of serological response to the first vaccination, which means that the serological response was not related to the vaccine reaction.The BNT162b2 vaccine is effective against symptomatic and severe COVID-19. In addition, none of the workers that acquired a SARS-CoV-2 infection after vaccination required hospitalization or medical care.Healthcare workers (HCWs) who had contracted COVID-19 before being vaccinated had a higher antibody response one week and three weeks after the first vaccine dose.

## Figures and Tables

**Figure 1 vaccines-10-02158-f001:**
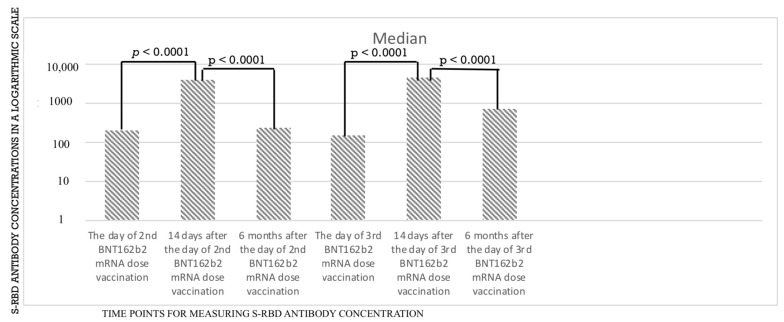
The change in anti S-RBD concentrations for the main group *n* = 170.

**Figure 2 vaccines-10-02158-f002:**
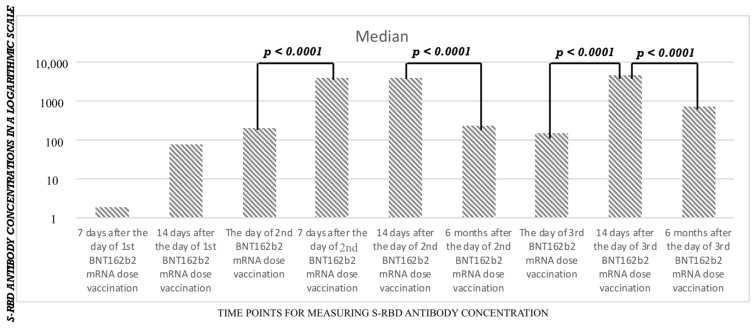
The dynamics of median anti SARS-CoV-2 S-RBD antibody concentrations measured in a subgroup of 49 healthcare workers.

**Table 1 vaccines-10-02158-t001:** The testing scheme for healthcare workers in the Hospital for Infectious Diseases in Warsaw, including both the main group and the subgroup.

Time of Sample Collection MAIN GROUP	SARS-CoV-2 IgG Antibodies	SARS CoV-2 IgM Antibodies	SARS-CoV-2 IgG S-RBD Antibodies
The day of the first vaccination	+	+	+
The day of the second BNT162b2 mRNA vaccination dose	+	+	+
14 days after the day of the second BNT162b2 mRNA vaccination dose	+	+	+
6 months after the day of the second BNT162b2 mRNA vaccination dose	+	-	+
The day of the third BNT162b2 mRNA vaccination dose	+	-	+
14 days after the day of the third BNT162b2 mRNA vaccination dose	+	-	+
6 months after the day of the third BNT162b2 mRNA vaccination dose	+	-	+
SUBGROUP			
7 days after the day of the first BNT162b2 mRNA vaccination dose	+	+	+
14 days after he day of the first BNT162b2 mRNA vaccination dose	+	+	+
7 days after the day of the second BNT162b2 mRNA vaccination dose	+	+	+

**Table 2 vaccines-10-02158-t002:** Positive results for SARS-CoV-2 IgM, IgG, and S-RBD antibody assessment among healthcare workers in the Hospital for Infectious Diseases in Warsaw.

	N	IgM Positive, *n* (%)	N	IgG Positive, *n* (%)	N	S-RBD Positive, *n* (%)
The day of 1st BNT162b2 mRNA dose vaccination, median [IQR]	211	11 (5.2)	221	45 (20.4)	203	54 (26.6)
The day of 2nd BNT162b2 mRNA dose vaccination, median [IQR]	46	19 (41.3)	115	22 (19.1)	114	111 (97.4)
14 days after 2nd BNT162b2 mRNA dose vaccination, median [IQR]	77	48 (62.3)	119	20 (16.8)	121	121 (100.0)
6 months after 2nd BNT162b2 mRNA dose vaccination, median [IQR]	10	1 (10.0)	205	34 (16.6)	209	209 (100.0)
The day of 3rd BNT162b2 mRNA dose vaccination, median [IQR]	N/A	N/A	187	34 (18.8)	184	184 (100.0)
14 days after 3rd BNT162b2 mRNA dose vaccination, median [IQR]	N/A	N/A	164	27 (16.5)	165	165 (100.0)
6 months after 3rd BNT162b2 mRNA dose vaccination, median [IQR]	N/A	N/A	169	63 (37.3)	170	170 (100.0)

N/A—not available.

**Table 3 vaccines-10-02158-t003:** Vaccine adverse effects after each dose of the BNT162b2 mRNA vaccine among the 49 healthcare workers in the subgroup, stratified by the level of response to the first dose of vaccine in the Hospital for Infectious Diseases in Warsaw.

Characteristic	N with Available Data	All	High Response N = 7	Low Response N = 42	*p* Value
**VAE after the first vaccine dose, *n* (%)**	47	32 (68.09)	6 (85.7)	26 (61.9)	0.403
**VAE after the second vaccine dose, *n* (%)**	42	35 (83.33)	5 (71.4)	30 (71.4)	1.000
**VAE after the third vaccine dose, *n* (%)**	40	37 (92.50)	6 (85.7)	31 (77.5)	0.447

**Table 4 vaccines-10-02158-t004:** The data for median and interquartile ranges of anti-S-RBD antibody titers in BAU/mL measured during the observation period in the group of 170 employees.

Time Point	S-RBD (in BAU/mL)
Day of the 2nd vaccine dose, median, [IQR]	226.600 [99.455–433.0]
14 days after the 2nd vaccine dose, median [IQR]	3338.2 [757.4–5791.0]
6 months after the 2nd vaccine dose, median [IQR]	224.9 [103.6–353.7]
Day of the 3rd vaccine dose, median [IQR]	170.2 [68.7–396.6]
14 days after the 3rd vaccine dose, median [IQR]	4525.8 [2804.9–7712.6]
6 months after the 3rd vaccine dose, median [IQR]	817.9 [402.2–3124.9]

**Table 5 vaccines-10-02158-t005:** The data for median and interquartile ranges of anti-S-RBD antibody titers in BAU/mL measured during the observation period in the group of 49 employees.

Time Point	S-RBD (in BAU/mL)
7 days after the 1st vaccine dose, median, [IQR]	1.85 [1.41–6.70]
14 days after the 1st vaccine dose, median [IQR]	77.51 [28.01–257.50]
Day of the 2nd vaccine dose, median [IQR]	198.90 [95.99–338.40]
7 days after the 2nd vaccine dose, median [IQR]	3846.80 [894.20–7309.50]
14 days after the 2nd vaccine dose, median [IQR]	3962.70 [2314.30–6465.10]
6 months after the 2nd vaccine dose, median [IQR]	232.10 [132.30–381.35]
Day of the 3rd vaccine dose	149.58 [77.16–388.62]
14 days after the 3rd vaccine dose, median [IQR]	4587.50 [3011.20–7478.00]
6 months after the 3rd vaccine dose, median [IQR]	701.05 [428.40–2265.6]

## Data Availability

The data presented in this study are available on request from the corresponding author.

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
