# Peer review of "Higher Immunological Response after BNT162b2 Vaccination among COVID-19 Convalescents—The Data from the Study among Healthcare Workers in an Infectious Diseases Center"

_vaccines, 2022, doi:10.3390/vaccines10122158_

Round 1

Reviewer 1 Report (New Reviewer)

Skrzat-KlapaczyÅ„ska et al. set out to characterize the antibody response to mRNA COVID-19 vaccination in health care workers exposed or not to SARS-CoV-2 infection. Even though, plenty of studies exist characterizing the antibody response to BNT162b2 vaccine and COVID-19 hybrid immunity, i.e., immune responses resulting from infection and immunity, has been already previously proposed as a provider of enhanced immunity, this paper does confirm previous findings and extends its geographic outreach. Nonetheless, there are serious flaws that must be addressed: 

  1. First, the authors start the abstract with the bold statement  “The BNT162b2 vaccination studies did not include groups that were heavily exposed to SARS-CoV-2 infection.” This sentence must be toned down, as vaccine studies were made in places of high SARS-CoV-2 incidence and since their approval, multiple studies have assessed BNT162b2 immune responses and protective effects in health care workers at multiple geographic locations. In line with this the sentence at the final of introduction “However, the BNT162b2 registration studies did not include groups that were particularly exposed to SARS-CoV-2 infection, including HCWs” must the rephrased and reference to previous studies with vaccine responses in HCW must be included. The focus of the study must not be that they are the first looking vaccine antibody responses into groups particularly exposed to SARS-CoV-2 infection, which is definitely not true, but that they are extending these findings with a new cohort. Just focusing on HCW as a higher exposed groups and setting aside the studies on long-term care facilities workers, the antibody response and protection of BNT162b2 vaccine has been established by the following papers, and thus references to these works should be contemplated: 

  • M. G. Thompson et al  MMWR Morb. Mortal. Wkly. Rep. 70, 2021.  

  • Benenson et al NJEM 2021 

  • Gonçalves et al Cell Rep Med 2021 

  • Hall et al Lancet 2021 

  • GiliRegev-Yochay The Lancet Regional Health - Europe 2021 

  • Salo et al Nat Commun 2022 

  • Spitzer et al JAMA 2022 

2-At the beginning of the Results section, it is stated that “In total 202 persons were recruited to the study, of which 170 persons were included in the final analysis.” The reasons for excluding the 32 participants, which corresponds to ~15.8% of initial recruited subjects must be stated 

3-Still in the results section the sentence “Most participants had COVID-19 either before being vaccinated (n=38, 22.6%), or after receiving the third BNT162b2 mRNA vaccine dose (n=45, 32.3%), eight (4.7%) persons had COVID-19 more than once.” must be reformulated. Adding the percentahes provided one arrives that about 50% of study participants had covid. Thus, “Most” is not an adequate term to describe this distribution. Replacing most by the actual percentage of study participants that actually had COVID-19 not only would be more exact but it would make for a clearer reading. 

4-Figure 1- the X-axis must be corrected, according to what is described in the methods and also to the information on the Y-axis, the authors did not determine anti-RBD titers. What is depicted is, again according to the authors methods, anti-RBD IgG antibody concentration in arbitrary units. In this line, any reference to antibody titers in the manuscript must be corrected to antibody concentration 

5- Still on Figure 1, the Y-axis needs to be labelled and, at minimum, a standard deviation of the anti-RBD IgG antibody concentration, reflecting interindividual variability, must be added. 

6-same applies to Figure 2, the X-axis must be corrected and anti-RBD IgG titers replaced by anti-RBD IgG concentration (AU), the Y-axis must be labelled, and, at minimum, the standard deviation must be included in the graph 

7-still on Figure 2 the label on the 4th bar must be corrected, it surely depicts 7 days after 2nd vaccine dose and not 7 days after 1st vaccine dose as it is described 

8-In order to support conclusion number 4, an additional figure and/or table is needed depicting the proportion of asymptomatic versus mild, moderate and severe SARS-CoV-2 infections in vaccinees that received 1, 2 or 3 vaccine doses. Comparisons with disease phenotypes distribution prior to vaccines should be included. 

In the discussion section 

9-all the mentions to antibody titers duly replaced by antibody concentrations. This is applicable all throughout the manuscript 

10-the reference 13 does not reflect the study nor the affirmation made by the authors and must be corrected. Additional references in vaccination studies made in HCW must be added (see initial remark) 

11-overall, the discussion should be rewritten and improved for content and depth but also for writing structure, only using paragraphs when needed, I.e., changing to a different subject 

In the conclusions section 

12-Point 4 of conclusions must be corrected, when the authors state “Healthcare workers (HCW) who had COVID-19 before being vaccinated had a higher immunological response one week and three weeks after the vaccination.” what they really mean is that COVID-19+ HCM had a higher antibody response, which is only one part of the immunological response. This sentence should also be completed, specifying after 1st vaccine dose. 

13-Still on point 4 “This may suggest that in a group of convalescents, only one vaccine dose may be enough to prevent symptomatic and severe COVID-19. However, further studies may be beneficial in providing better support for this conclusion.” This assumption is not supported by the literature and should be nuanced accordingly, if not removed in its entirety.

Author Response

Skrzat-KlapaczyÅ„ska et al. set out to characterize the antibody response to mRNA COVID-19 vaccination in health care workers exposed or not to SARS-CoV-2 infection. Even though, plenty of studies exist characterizing the antibody response to BNT162b2 vaccine and COVID-19 hybrid immunity, i.e., immune responses resulting from infection and immunity, has been already previously proposed as a provider of enhanced immunity, this paper does confirm previous findings and extends its geographic outreach. Nonetheless, there are serious flaws that must be addressed: 

  1. First, the authors start the abstract with the bold statement  “The BNT162b2 vaccination studies did not include groups that were heavily exposed to SARS-CoV-2 infection.” This sentence must be toned down, as vaccine studies were made in places of high SARS-CoV-2 incidence and since their approval, multiple studies have assessed BNT162b2 immune responses and protective effects in health care workers at multiple geographic locations.

We toned down the statement and now it is: “The BNT162b2 vaccination studies did not specifically focused on groups that were heavily exposed to SARS-CoV-2 infection”

In line with this the sentence at the final of introduction “However, the BNT162b2 registration studies did not include groups that were particularly exposed to SARS-CoV-2 infection, including HCWs” must the rephrased and reference to previous studies with vaccine responses in HCW must be included. The focus of the study must not be that they are the first looking vaccine antibody responses into groups particularly exposed to SARS-CoV-2 infection, which is definitely not true, but that they are extending these findings with a new cohort. Just focusing on HCW as a higher exposed groups and setting aside the studies on long-term care facilities workers, the antibody response and protection of BNT162b2 vaccine has been established by the following papers, and thus references to these works should be contemplated: 

  • G. Thompson et al  MMWR Morb. Mortal. Wkly. Rep. 70, 2021. 
  • Benenson et al NJEM 2021
  • Gonçalves et al Cell Rep Med 2021
  • Hall et al Lancet 2021
  • GiliRegev-Yochay The Lancet Regional Health - Europe 2021
  • Salo et al Nat Commun 2022
  • Spitzer et al JAMA 2022

This has been rephrased and the references were added.

2-At the beginning of the Results section, it is stated that “In total 202 persons were recruited to the study, of which 170 persons were included in the final analysis.” The reasons for excluding the 32 participants, which corresponds to ~15.8% of initial recruited subjects must be stated 

It has been explained. “Thirty two study participants were excluded from the analysis if at least one blood sample was missing from the study scheme”

3-Still in the results section the sentence “Most participants had COVID-19 either before being vaccinated (n=38, 22.6%), or after receiving the third BNT162b2 mRNA vaccine dose (n=45, 32.3%), eight (4.7%) persons had COVID-19 more than once.” must be reformulated. Adding the percentahes provided one arrives that about 50% of study participants had covid. Thus, “Most” is not an adequate term to describe this distribution. Replacing “most” by the actual percentage of study participants that actually had COVID-19 not only would be more exact but it would make for a clearer reading. 

We reformulated this sentence in the results section and also in the abstract. The beginning of this sentence is now: ).  “More than one fifth of participants had COVID-19 either before being vaccinated (n=38, 22.6%),”

4-Figure 1- the X-axis must be corrected, according to what is described in the methods and also to the information on the Y-axis, the authors did not determine anti-RBD titers. What is depicted is, again according to the authors methods, anti-RBD IgG antibody concentration in arbitrary units. In this line, any reference to antibody titers in the manuscript must be corrected to antibody concentration 

Thank you for this remark. We have provided the S-RBD antibody concentrations in a logarithmic scale in order to clarify these changes. We’ve also corrected every reference to antibody titers to antibody concentration

5- Still on Figure 1, the Y-axis needs to be labelled and, at minimum, a standard deviation of the anti-RBD IgG antibody concentration, reflecting interindividual variability, must be added. 

Thank you for this remark. We have labeled the y-axis. However, the standard deviation shows the dispersion of results when the mean value is provided. We believe that better suited for our results dispersion where the median values of antibody titers are shown would be the interquartile ranges, and yet we have provided them accordingly.

6-same applies to Figure 2, the X-axis must be corrected and anti-RBD IgG titers replaced by anti-RBD IgG concentration (AU), the Y-axis must be labelled, and, at minimum, the standard deviation must be included in the graph 

Thank you for this remark. The X-axis shows the  S-RBD antibody titers in a logarithmic scale in order to clarify these changes. Therefore, we clarified the Figure’s description. In addition, the standard deviation shows the dispersion of results when the mean value is provided. We believe that better suited for our results dispersion where the median values of antibody titers are shown would be the interquartile ranges, and yet we have provided them accordingly.

7-still on Figure 2 the label on the 4th bar must be corrected, it surely depicts 7 days after 2nd vaccine dose and not 7 days after 1st vaccine dose as it is described 

It has been corrected

8-In order to support conclusion number 4, an additional figure and/or table is needed depicting the proportion of asymptomatic versus mild, moderate and severe SARS-CoV-2 infections in vaccinees that received 1, 2 or 3 vaccine doses. Comparisons with disease phenotypes distribution prior to vaccines should be included. 

According to the point 12 suggestion – the conclusion 4 has been corrected  and now it is: “Healthcare workers (HCW) who had COVID-19 before being vaccinated had a higher antibody response one week and three weeks after the first vaccine dose.”

In the discussion section 

9-all the mentions to antibody titers duly replaced by antibody concentrations. This is applicable all throughout the manuscript

It has been corrected 

10-the reference 13 does not reflect the study nor the affirmation made by the authors and must be corrected. Additional references in vaccination studies made in HCW must be added (see initial remark) 

The reference 13 has been corrected. Additional references have been added

11-overall, the discussion should be rewritten and improved for content and depth but also for writing structure, only using paragraphs when needed, I.e., changing to a different subject 

The discussion has been rewritten according to reviewer’s suggestion

In the conclusions section 

12-Point 4 of conclusions must be corrected, when the authors state “Healthcare workers (HCW) who had COVID-19 before being vaccinated had a higher immunological response one week and three weeks after the vaccination.” what they really mean is that COVID-19+ HCM had a higher antibody response, which is only one part of the immunological response. This sentence should also be completed, specifying after 1st vaccine dose. 

This has been corrected.

13-Still on point 4 “This may suggest that in a group of convalescents, only one vaccine dose may be enough to prevent symptomatic and severe COVID-19. However, further studies may be beneficial in providing better support for this conclusion.” This assumption is not supported by the literature and should be nuanced accordingly, if not removed in its entirety.

We agree that this assumption is not supported by the literature and it has been removed

Reviewer 2 Report (New Reviewer)

This is a serological follow-up of a group of infectious disease hospital workers, mostly nurses after COVID vaccination. In this cohort of 170 subjects several blood specimens were taken and COVID antibodies were measured using total IgM and IgG, as well as S-RBD antibody measurement. Also, vaccine associated adverse events were recorded. Altogether this was a closely monitored cohort which can yield some detailed information on the antibody responses and vaccine reactions after Pfizer/BioNTech’s COVID-19 vaccine.

Although this is a closed cohort with many blood collections it still is mainly confirmatory for its results. Much of the same has been previously reported, partly by the same authors.

The main point presented in the heading “Higher immunological response after COVID vaccination among COVID-19 convalescents” is based on a very small number of subjects. In the subgroup analysis, the authors state that a very high immunological response was seen in 6/7 COVID-19 convalescents vs. 1/42 non convalescents. Six out seven is not much. Two questions follow.

aa.   What is a very high response? It is not defined anywhere.

b.  Why was this comparison not done for all 38 subjects that according to Table 2 were convalescents before 1st vaccination

The responses should be presented as GMTs and comparing the 38 convalescents with the 122 non-convalescents after the first dose. The same could be then done with convalescents before the 2nd dose.

The paper is a bit long and presents data that are not necessary. Vaccine adverse events are presented in the text and Table 3. Of these Table 3 is redundant and could be deleted. The demographics is presented in the text and Table 2. Again Table 2 is redundant. “High” and “low” response in Table 2 is not defined. GMTs should be used instead.

In summary, the paper should be revised taken into consideration the above points. In addition, the paper should be condensed by deleting Tables 2 and 3.

Author Response

This is a serological follow-up of a group of infectious disease hospital workers, mostly nurses after COVID vaccination. In this cohort of 170 subjects several blood specimens were taken and COVID antibodies were measured using total IgM and IgG, as well as S-RBD antibody measurement. Also, vaccine associated adverse events were recorded. Altogether this was a closely monitored cohort which can yield some detailed information on the antibody responses and vaccine reactions after Pfizer/BioNTech’s COVID-19 vaccine.

Although this is a closed cohort with many blood collections it still is mainly confirmatory for its results. Much of the same has been previously reported, partly by the same authors.

The main point presented in the heading “Higher immunological response after COVID vaccination among COVID-19 convalescents” is based on a very small number of subjects. In the subgroup analysis, the authors state that a very high immunological response was seen in 6/7 COVID-19 convalescents vs. 1/42 non convalescents. Six out seven is not much. Two questions follow.

  1. What is a very high response? It is not defined anywhere.

There is a paragraph in the methods section: “In the statistical analyses persons with high and low response on the six months after the third vaccine dose were compared. High response was defined as values over 433 BAU/mL based on other studies (12).” We changed everywhere the term “very high response” to “high response” in the manuscript .

  1. Why was this comparison not done for all 38 subjects that according to Table 2 were convalescents before 1stvaccination

The responses should be presented as GMTs and comparing the 38 convalescents with the 122 non-convalescents after the first dose. The same could be then done with convalescents before the 2nd dose.

Thank you for this remark. According to the reviewer’s suggestion Table 2 was removed from the manuscript. However, this subgroup analysis was focused on vaccine’s booster dose efficacy after 6 months. Therefore, we stratified the available HCWs antibody titers data on high and low response. However, we believe that reviewer’s suggestions is very valuable and worth considering for future analysis.

The paper is a bit long and presents data that are not necessary. Vaccine adverse events are presented in the text and Table 3. Of these Table 3 is redundant and could be deleted. The demographics is presented in the text and Table 2. Again Table 2 is redundant. “High” and “low” response in Table 2 is not defined. GMTs should be used instead.

Table 2 and Table 3 have been removed according to the suggestions High response was defined in the methods section “In the statistical analyses persons with high and low response on the six months after the third vaccine dose were compared. High response was defined as values over 433 BAU/mL based on other studies (12).”

In summary, the paper should be revised taken into consideration the above points. In addition, the paper should be condensed by deleting Tables 2 and 3.

Table 2 and Table 3 have been removed according to the suggestions

Round 2

Reviewer 1 Report (New Reviewer)

Even though the authors have indeed improved the manuscript in response to revision, there are still two main issues that ought to be rectified.

1-The number of studies reporting COVID-19 mRNA vaccine responses in HCW is immense. Of these, 7 studies have been suggested to the authors but only 3 were included in the revised submission. The inclusion of only 3 references fails to reflect the extension of work previously done on the subject.

2- The authors mention that they would rather state the median and the IQR of the anti-RBD IgG concentration in their study population oven the mean plus standard deviation. Nonetheless, I could not see any mention to either the median or IQR in the revised manuscript. These data ought to be included.

Author Response

Reviewer 1

1-The number of studies reporting COVID-19 mRNA vaccine responses in HCW is immense. Of these, 7 studies have been suggested to the authors but only 3 were included in the revised submission. The inclusion of only 3 references fails to reflect the extension of work previously done on the subject.

This has been corrected. All suggested references have been added.

2- The authors mention that they would rather state the median and the IQR of the anti-RBD IgG concentration in their study population oven the mean plus standard deviation. Nonetheless, I could not see any mention to either the median or IQR in the revised manuscript. These data ought to be included.

Thank you for this remark. Due to some technical problems our supplementary file could not be uploaded with our revisions. We are sorry for the inconvenience. Here we provide a supplementary file of Tables 4 and 5 with data for the anti-S-RBD antibody titers in medians and interquartile ranges. However, in order to achieve more clarity in Figures 1 and 2 we have decided to present only median values in a logarithmic scale. 

Supplementary File: Table 4 and 5 with provided data for median and intrerquartile ranges of anti-S-RBD antibody titers in BAU/mL measured during the observation period in the group of 170, and 49 employees, respectively.

Table 4.

S-RBD (in BAU/mL)

Day of the 2nd vaccine dose, median, [IQR]

226.600 [99.455–433.0]

14 days after the 2nd vaccine dose, median [IQR]

3338.2 [757.4–5791.0]

6 months after the 2nd vaccine dose, median [IQR]

224.9 [103.6–353.7]

Day of the 3rd vaccine dose, median [IQR]

170.2 [68.7–396.6]

14 days after the 3rd vaccine dose, median [IQR]

4525.8 [2804.9–7712.6]

6 months after the 3rd vaccine dose, median [IQR]

817.9 [402.2–3124.9]

Table 5.

S-RBD (in BAU/mL)

7 days after the 1st vaccine dose , median, [IQR]

1.85 [1.41–6.70]

14 days after the 1st vaccine dose, median [IQR]

77.51 [28.01–257.50]

Day of the 2nd vaccine dose, median [IQR]

198.90 [95.99–338.40]

7 days after the 2nd vaccine dose, median [IQR]

3846.80 [894.20–7309.50]

14 days after the 2nd vaccine dose, median [IQR]

3962.70 [2314.30–6465.10]

6 months after the 2nd vaccine dose, median [IQR]

232.10 [132.30–381.35]

Day of the 3rd vaccine dose

149.58 [77.16–388.62]

14 days after the 3rd vaccine dose, median [IQR]

4587.50 [3011.20–7478.00]

6 months after the 3rd vaccine dose, median [IQR]

701.05 [428.40–2265.6]

Reviewer 2 Report (New Reviewer)

Accept

Author Response

Accept

Thank you

Round 3

Reviewer 1 Report (New Reviewer)

I have nothing else to add.

This manuscript is a resubmission of an earlier submission. The following is a list of the peer review reports and author responses from that submission.

Round 1

Reviewer 1 Report

The manuscript entitled “Is One Dose of COVID-19 Vaccine Enough Among COVID-19 Covalescents? – The Data From The 2 Study Among Healthcare Workers in an Infectious Diseases Center” by Skrzat-KlapaczyÅ„ska et al. sent for publication to the IJERPH, is focusing on a very important nowadays topic – vaccination against COVID-19 disease. However, the authors present a very limited characterization of the immune response after vaccination and the experiments did not provide sufficient information to support their theory. The title of the presented manuscript largely matches the title of another article: Focosi D, Baj A, Maggi F. Is a single COVID-19 vaccine dose enough in convalescents ? Hum Vaccin Immunother. 2021 Sep 2;17(9):2959-2961. doi: 10.1080/21645515.2021.1917238. Epub 2021 May 5. PMID: 33950788; PMCID: PMC8108188.

A careful revision of the English language is strictly necessary to correct grammatical errors and improve the level of standard English. Indeed, in the present form, many parts of the manuscript are not understandable due to poor use of the English language. Please, consider the academic proofreading service or at least the help of an English native speaker.

The research is not designed properly.

In the study, I do not see a clear division of the examined health workers into two distinguished groups, those who have Covid-19 and have developed anti-SARS-CoV-2 Spike and anti-SARS CoV-2 Nucleocapsid  Ab, and those who have not passed Covid-19 (do not have  anti-SARS-CoV-2 Spike and anti-SARS CoV-2 Nucleocapsid  Ab).

I would like to see the specific anti-RBD IgG (kAU/ml) wild type compared to the specific IgG induced by the vaccination.

My advice to the authors of the manuscript is to rewrite their article completely and to obtain additional data that show the difference in the immune response after vaccination of the target groups (relapsed and vaccinated and non-relapsed and vaccinated) and resubmit. 

Reviewer 2 Report

It is an interesting study that requires attention to important flaws in the exposition of methodology, results, and conclusions. Also, it contains several misspellings, e.g., “covalescents”, “dicrease”, and “emploees”.

Title. The idea of "one dose" expressed in the title is not clear since there are vaccines such as Johnson & Johnson’s 's whose complete schedule is 1 dose.  Perhaps the authors meant "a booster dose". Additionally, the objective is limited to the Pfizer vaccine, which is not reflected in the title.

Abstract. It requires adjustments according to the comments that are exposed later.

Introduction. There is a lack of information on studies conducted in the convalescent population, which should be the focus of the study according to the title. The objective does not match the title.

Material and methods. Specify the study design, sampling technique and sample size calculation. Definition of variables: How was "enough" defined (title says: Is one dose of covid-19 vaccine "enough" among covid-19 convalescents). Was it based on the levels of SARS CoV-2 IgM antibodies, SARS-CoV-2 IgG antibodies? reduction in the incidence of covid cases, and/or reduction in the incidence of severe cases? How was safety measured? and efficacy? Was the study design a randomized controlled clinical trial? (this design is required for medicating efficacy)

Results.

Table 2, it is redundant. It says “The study participants’ baseline characteristics at baseline”

Table 2, subheading of column 2 please use “number of employees” and not “number of patients”

Table 2, what does “High” and “low” response 6 months after the third dose means? What does the * mean?

Figure 1. What does “change” in anti S-RBD titers mean

The authors must focus the results on covid-19 convalescents with and without a booster dose to be consistent with what the title seems to say

Discussion. What “high level” of efficacy means?

Discussion and conclusions. The authors must focus and answer what the title seems to say.